# DIVIDE-AND-CLUSTER: SPATIAL DECOMPOSITION BASED HIERARCHICAL CLUSTERING

## ABSTRACT

This paper is about increasing the computational efficiency of clustering algorithms. Many clustering algorithms are based on properties of relative locations of points, globally or locally, e.g., interpoint distances and nearest neighbor distances. This amounts to using less than the full dimensionality $D$ of the space in which the points are embedded. We present a clustering algorithm, Divide-and-Cluster (DAC), which detects local clusters in small neighborhoods and then merges adjacent clusters hierarchically, following the Divide-and-Conquer paradigm. This significantly reduces computation time which may otherwise grow nonlinearly in the number $n$ of points. We define local clusters as those within hypercubes in a recursive hypercubical decomposition of space, represented by a tree. Clusters within each hypercube at a tree level are merged with those from neighboring hypercubes to form clusters of the parent hypercube at the next level. We expect DAC to perform better than many other algorithms because (a) as clusters merge into larger clusters (components), their number steadily decreases vs the number of points, and we cluster (b) only neighboring (c) components (not points). The recursive merging yields a cluster hierarchy (tree). Further, our use of small neighborhoods allows piecewise uniform approximation of large, nonuniform, arbitrary shaped clusters, thus avoiding the need for global cluster models. We present DAC's complexity and experimentally verify the correctness of detected clusters on several datasets, posing a variety of challenges, and show that DAC's runtime is significantly better than representative algorithms of other types, for increasing values of $n$ and $D$.

## 1 INTRODUCTION

Finding clusters formed by $n$ data points in a $D$-dimensional space is a very frequent operation in machine learning, even more so in unsupervised learning. This paper is about increasing the computational efficiency of clustering. The notion of cluster is fundamentally a perceptual one; its precise definition, used implicitly or explicitly by clustering algorithms, varies. Given a set of n points $x_1, x_2, ..., x_n$ in a $D$-dimensional space ($x_i \in R^D$), clustering is aimed at identifying different groupings/components of nearby points according to the definition of cluster being used. In this paper, we model a cluster as a contiguous overlapping set of small neighborhoods such that: points in each neighborhood are distributed (nearly) uniformly, the cluster shape is arbitrary, and at their closest approach any two clusters are separated by a distance larger than the distance between nearby within-cluster neighbors. Many common clustering algorithms are based on assumptions about global properties of point positions, such as distances between points or sets of points, as in K-means. Another class of algorithms uses local structure, defining locality via edge-connectivity in different types of graphs defined by the points. Other algorithms use nearby points, e.g., located within a chosen size neighborhood. There are a few approaches that make full use of the $D$-dimensional geometry, e.g., those based on Voronoi neighborhoods (Ahuja & Tuceryan, 1989); while they work well, their constituent algorithms (e.g., for Voronoi tessellation) are available for only small values of $D$.

In this paper, we present a clustering algorithm, called Divide-and-Cluster (DAC), that detects clusters in small local neighborhoods formed by hierarchically decomposing the $D$-space and then re-

peatedly grows them outwards, by remerging chopped parts of a cluster stretching across neighborhood boundaries. This implements the Divide-and-Conquer paradigm of computation. We use hypercubical neighborhoods formed by a recursive hypercubical tessellation of the $D$-space, represented by a tree. Clusters straddling boundaries of neighboring hypercubes/nodes at one level are combined to form clusters of their common parent node at the next level, in a bottom up tree traversal. Once the recursive merger is completed at the tree root level, all detected clusters are agglomeratively merged, closest clusters first, to form a cluster hierarchy.

We evaluate our algorithm against other algorithms in the following ways. (1) Correctness of Detected Clusters: We compare detected clusters in the standard, Fundamental Clustering Problems Suite of 2D and 3D datasets (FCPS), which poses a variety of important clustering challenges. (2) Runtimes: We compare runtimes on datasets having a range of $n$ and $D$ values. (3) Downstream Task: We observe the impact of DAC as a clustering substitute in a CNN.

The advantages of the proposed algorithm lead to the following contributions:

**1. Spatial Divide-and-Conquer:** Our approach recursively divides the $D$ space in a using a fixed, predetermined geometric criterion. This helps divide the points without using additional computation that may be needed to evaluate the division criteria, The resulting logarithmic run-time increases with $E$. DAC also offers a new mechanism for trade-off among $n$, $D$ and $E$.

**2. Point Components:** DAC's time complexity over global/points based algorithms further benefits from the use of components of points instead of individual points: (a) It limits computation of closest distance between neighboring components to mutually visible points (e.g., half) of the component boundaries (instead of all points in the components), and (b) the agglomerative generation of hierarchical clusters need only merge entire, closest components, implying a complexity in terms of the smaller number of components vs points.

**3. Cluster Shape:** DAC's local cluster detection allows arbitrarily simple local models of clusters, avoiding cluster wide global models. For example, by repeated division, DAC can increasingly improve the validity of piecewise uniform approximation of cluster density.

**4. Cluster Density:** Compositional detection of a cluster from its contiguous small pieces also allows detection of clusters of arbitrary shapes, thus avoiding shape models such as isotropy/compactness used by K-means.

In the following, we first review related work (Sec 2), followed by spatial decomposition (tree representation) used to define DAC (Sec 3), DAC algorithm (Sec 4), computational complexity (Sec 5), experimental results (Sec 6), and limitations and conclusions (Sec 7).

## 2 RELATED WORK

Clustering obtains a partition of the points that minimizes a cost function chosen to reflect the types of clusters to be detected, often along with some user specified parameters. An example of a cost function is the Sum of Squared Error (SSE), the sum of the squares of distances of points from their cluster centers. Hierarchical clustering approaches derive an agglomerative tree (hierarchy) of clusters formed by repeatedly merging, bottom up, lowest cost first. Partitioning algorithms form the hierarchy by repeatedly splitting the clusters, maximally reducing the cost at each step. Following are some broad categories of algorithms:

**Graph-based clustering** algorithms typically use a partitioning strategy on graph of data points with weighted edges. Examples are Minimal Spanning Tree (MST) and Shared Nearest Neighbor (SNN) clustering algorithms. MST defines edge weight as distance between two points and removes $k - 1$ highest weighted edges from the MST of the graph to obtain $k$ clusters, where $k$ is a parameter. SNN defined similarity of two points as threshold overlap in nearest neighbors (where $k'$ nearest neighbors are found for each point), similar vertices are connected, and connected components are discovered to return the clusters.

**Spectral approaches** aim to partition a graph such that similar points are together and dissimilar are not. It defines a similarity matrix that contains similarity score for each pair of points, and uses this matrix to find the graph cut with minimum cost.

**Density-based clustering algorithms** start with some core clusters (e.g., individual points) and gradually grow each cluster by including proximal points until a growth criterion allows. Examples include DBSCAN (Ester et al., 1996), OPTICS (Ankerst et al., 1999) and Mean shift based (Cheng, 1995) clustering. DBSCAN defines a core cluster as consisting of those points that have more than a minimum number minpoints of points in their $\in$-neighbourhoods. Each core cluster grows by including all points lying in $\in$-neighbourhoods of its current points, and repeating this process until no growth is possible. The process then repeats from a new unclustered point until all data points are clustered.

**Mesh-based clustering algorithms** quantize the data space into small units (mesh cells) and aggregate all points within one mesh cell into one indivisible unit. This may be interpreted as working on a coarse resolution, where the algorithm is unable to distinguish between points in the same mesh cell. Then, all computation is performed on the mesh cells rather than the data points. The time complexity of the operations then is a function of the number of mesh cells, not the number of data points. This presents a trade-off between the cluster quality (coarser meshes would produce poorer clustering) and the time required to compute the clusters (proportional to number of mesh cells). Some examples are STING (Wang et al., 1997), CLIQUE (Agrawal et al., 2005) and WaveCluster (Sheikholeslami et al., 1998).

Approximate clustering algorithms offer trade-offs between complexity and the degree to which the detected clusters satisfy the clustering criteria. Other algorithms focus on hierarchical, single linkage clustering. Yet others present variations of tree representations but not directly impacting or related to the algorithm we propose. Examples include Curtin et al. (2013); Abboud et al. (2019); Liu et al. (1970); Borůvka (1926). Because of this difference in their objective, we do not consider them any further in this paper.

## 3 TREE REPRESENTATION

For concreteness, we first discuss the case of $D = 3$, the octree representation (Shi & Malik, 2000; Jackins & Tanimoto, 1980; Meagher, 1982; Zahn, 1971; VanderPlas, 2016; Liao et al., 2004) of the occupancy of space by non-point objects, obtained using recursive decomposition of the space into octants. Given an object, we start with the entire space as a single (starting) cube (block). At any stage, if a block under consideration is completely contained within the object it is left alone; otherwise, it is divided into its eight octants which are added to a pool of blocks to be processed; this procedure is repeated for each unprocessed block until the pool is empty (all blocks are either completely within or completely outside the object, or they are of the smallest, specified (precision/resolution level) mesh cell size.

This recursive subdivision yields an octree description of the occupancy of 3D space. Each block corresponds to a node in the tree. The root (level 0) of the octree represents the largest size block, representing the limit of object size that can be represented. Successive levels represent blocks whose side lengths halve with each lower level, as does the separation between adjacent blocks. We denote the side length of the smallest block, the mesh cell, by unity. For the largest (universe) hypervolume $[0, E]^D$, the maximum number of levels $L$ then is $\log_2 E$. An octree may also have fewer than $L$ levels if all object blocks turn out to be of side lengths larger than unity; the nodes corresponding to these larger blocks form octree's leaves and may occur at any of the $L$ levels. We label a leaf node black/white if its block is completely contained within the object/free space. If a node is neither black nor white, then it is labelled gray, meaning it has both black and white descendants. Black nodes not contained inside other black nodes are termed 'maximal' black nodes.

To generalize to the case of points in the $D$-dimensional space, we recursively divide the space using $D$-dimensional hypercubes. Each dimension '$x$' of the universe hypervolume $[0, E]^D$ is split into two halves by the half plane $x = E/2$. This partitions each $D$-dimensional block into $2^D$ children hypercubes, each of half the edge length, and $1/2^D$ times the volume, of the parent block, yielding a hypertree of order $2^D$. To associate a child node's id with one of its $2^D$ possible positions $r_i(r_i = 0, 1, 2, \cdots, 2^D - 1)$, we use the binary representation of $r_i$ itself as the node's id. We associate the lower/upper half of each dimension with 0/1. This maps all the $2^D$ children to unique positions inside the parent (Fig 1(a)).

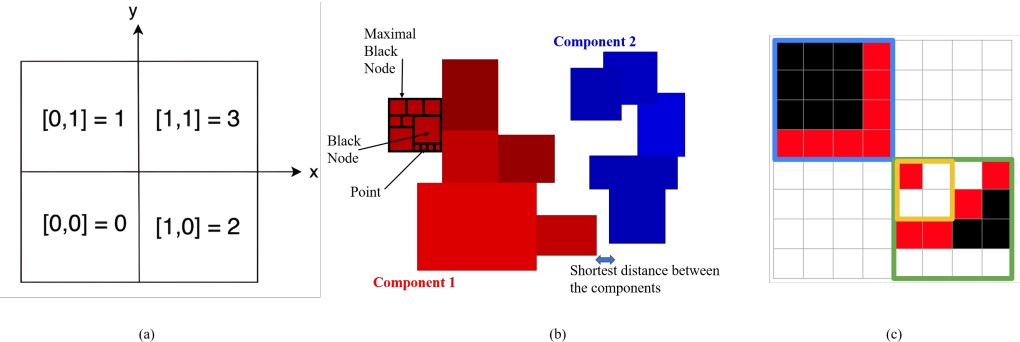

Figure 1: Overview: (a) A parent node's decomposition into its $2^D$ children. A child node's id is a binary number in which each bit corresponds to one dimension and is 0(1) if the child position is in the lower(upper) half in that direction. For example, the bottom right child's id is 10, being in the upper half along x-axis and lower along y-axis. (b) An illustration of the major steps in DAC via an image. The smallest squares stand for mesh cells. Black nodes are the largest hypercubes filled with black mesh cells. Neighboring Black nodes (nonsiblings in hypertree) concatenate to form the largest hypercube or Maximal Black Node (MBN) and MBNs concatenate to form the largest black region, or component - corresponding to a dense cluster (Component 1). During hierarchical clustering, components merge in the order of increasing intercomponent distance. (c) Boundary construction. The mesh cells in red constitute mutually visible boundary between blue and green nodes. Only these cells are involved in computing distance between the two nodes. The yellow node has holes that expose mesh cells in the center of green node to the green visible boundary.

## 4 ALGORITHM

### 4.1 INSERTION OF POINTS INTO THE HYPERTREE

The first step is to associate each data point with the smallest block (mesh cell or unit hypercube) or leaf nodes of the hypertree. Points are inserted in a top-down traversal of the tree to the leaves. Since storing all $2^D$ children of non-leaf nodes is infeasible for large $D$, and DAC requires only black grey (non-white) nodes, we explicitly store only these nodes. This can be achieved using, e.g., a Linked List of non-white child nodes, or a Hash Map of (key, value) pairs with $r \in \{0, 1, 2, 2^D - 1\}$ as key and the child node as value, respectively. We use the latter as it has a faster look up time.

### 4.2 FINDING MAXIMAL BLACK NODES (MBN)

We call the black nodes not contained in any other black nodes as Maximal Black Nodes (MBN). They may occur at the leaf level or at an interior level. The latter are identified in a depth first traversal, by coloring every parent black if all its children are black (Appendix A.1 - procedure *set color*). and then deleting all the children (Appendix A.2 - procedure *find maximal black nodes*). MBN are represented by $(\gamma_1, ..., \gamma_{n_b})$, where $(n_b)$ denotes the number of maximal black nodes (Fig 1(b)).

### 4.3 COMPUTING THE ADJACENCY MATRIX OF MBN

MBN adjacency matrix holds the shortest distance between each pair of MBN, which is the shortest distance between a pair of mesh cells in the two MBN. To compute these distances, we exploit the fact that these mesh cells must lie on the mutually visible boundaries of the nodes.

The shortest distance between two MBN is given by the straight line distance between two mutually visible (nonmaximal) black nodes at the boundaries of the MBN located. To identify these boundary node pairs, we start at the MBN and move up the hypertree. At each next level, we use the boundaries of sibling nodes to update the adjacency matrix of the two MBN and compose the parent node boundary using the sibling node boundaries. Fig 1(b) illustrates these steps, which are also given in the following top-down, recursive *fill matrix* procedure:

**1. Traverse** the tree to access the leaves.

**2. Adjacency Matrix:** From each leaf, traverse bottom up to recursively compute the adjacency matrix values for the parent node using the mutually visible boundaries of all pairs of a parent node's non-empty children. For a pair of sibling nodes $\{s_1, s_2\}$, their visible boundary is computed as the union of their mutually visible pairs of faces. A sibling's face is visible if it is in those dimensions (axes, directions) along which the sibling is shifted. In the binary representation of a child hypercube, a 0/1 represents the child's position coordinate along a dimension being in the lower/higher half along that dimension; if the siblings $\{s_1, s_2\}$ have coordinates 0 and 1, respectively, the visible boundary would include the higher boundary face of $\{s_1\}$ and the lower boundary face of $\{s_2\}$ The union of these face pairs from all directions constitutes the visible boundary shared between the two siblings. Then the distance between the siblings is the minimum of the distances for each pair of cross-sibling mesh cells on the visible parts of the boundaries. If less than the current entry, the adjacency matrix is updated (Fig 1(b)).

**3. Boundary Construction:** A node boundary consists of mesh cells which are visible when viewing the node from along positive and negative along each axis. A node's boundary is composed from the appropriate boundaries of each of its children. For example in 2D (Fig 1(c)), the boundary of the bottom-left child $B_1$ may extend the bottom left boundary of the parent . While doing so, the boundary of a sibling hidden behind each child node (say $B_2$) also need to be checked in case mesh cells from $B_2$ are exposed through any holes in $B_1$. Starting at the root of the hypertree, this recursive boundary construction ends at a black node by returning its boundary and no updates to the adjacency matrix (Appendix A.4 - procedure *get node boundary*). Appendix A.4 procedure *get node boundary* gives details.

### 4.4 Creating components by merging MBN

This step identifies neighboring components, formed by MBN separated by unit distance (i.e., pixels in one MBN have neigbhoring pixels in the other), represented by $(\zeta_1, \zeta_2, ..., \zeta_{n_c})$. A component is the largest set of MBN such that any two MBN can be reached from each other following a chain of neighboring MBN. An MBN thus represents a point cluster split across different hypertree nodes, corresponding to neighboring hypercubes. This reduces the original problem of clustering points into one of clustering components. Since the number of components is usually much smaller that the number of points, it reduces the time required to extract clusters (Sec 5).

To find the neighboring MBN, we locate unit distance entries in the MBN adjacent matrix, and merge the corresponding MBN using a union find algorithm. This yields an identifier $f(.)$ for each MBN. Two MBN belonging to the same component have the same $f(.)$ value. The adjacency matrix is updated to hold the pairwise distance between components: each entry $(i, j)$ in the original matrix is updated with $(f(i), f(j))$, giving a new, component adjacency matrix $(n_c \times n_c)$.

### 4.5 Component clustering

The components obtained are embedded into a graph as vertex nodes. The weight of the edge connecting two vertices is the shortest distance between their mesh cells, which must be for two mesh cells lying on mutually visible boundaries of the components. Hierarchical clustering can then be done using a graph based clustering algorithm. We use the MST clustering algorithm. We first compute the MST of the component graph to form a single component (cluster) connecting all nodes with minimum length (weight) edges. Then, as in Zahn (1971), we remove the longest edge to create two components. By repeatedly removing an edge with the largest weight, we repeatedly increase the component (cluster) count by one. This yields a cluster tree, from which any desired number of clusters $k$ may be obtained by removing the $k - 1$ longest edges, along with their adjacency matrix. This algorithm amounts to a single linkage hierarchical clustering algorithm, but the links are between MBN (clusters of points) and not points themselves, and therefore usually much smaller in number.

## 5 Time complexity

The algorithm consists of the procedures:

**1. Insertion of points into the hypertree:** Each of the $n$ data points is inserted at each of the $log_2(E)$ tree levels, resulting in a time complexity of $O(n \cdot \log E)$. Storing each data point and non-white nodes (neglecting the smaller needs, e.g., for hash storage) leads to a space complexity of $O(n + N_{nw})$, where $N_{nw}$ indicates the number of non-white nodes in the hypertree.

**2. Coloring (*set color* procedure):** As each black node is accessed a recursive call is made to color each non-white child. Hence, the time complexity is $O(N_{nw})$. Since the color has to be stored for each non-white node the space requirement is $O(N_{nw})$.

**3. Marking maximal black nodes (*find maximal black nodes* procedure):** This step involves depth first traversal of the tree to extract maximal black nodes and to associate them with their children nodes. Hence both time and space complexities are $O(N_{nw})$.

**4. Computing the adjacency matrix (*fill matrix* procedure):** This step consists of two procedures

**(a)** *Computing distance between points on the visible boundary:* This step computes the distance between pairs of mutually visible mesh cells on the boundaries of the two sibling nodes and updates the sibling adjacency matrix Hence the time complexity is $O(SA^2)$, where $SA$ denotes the surface area of the sibling nodes. To store the visible boundaries requires $O(SA)$ space.

**(b)** *Boundary construction:* The time and space required to insert mesh cells from child nodes into the parent's boundary is $O(SA$ of the parent boundary$)$. However, this is dominated by the identification of mesh cells exposed to the boundary via holes in its neighbours, which grows with the child nodes surface area as $O(SA \cdot \log(SA))$(Appendix 4b).

**5. Merging maximal black nodes into components:** This step consists of two procedures

**(a)** *Finding maximal black nodes at a unit distance from each other:* This requires traversing the adjacency matrix and has time complexity $O(n_b^2)$, $n_b$ being the number of maximal black nodes.

**(b)** *Merging proximal maximal black nodes into components:* This requires running a union find algorithm over the maximal black nodes and has time complexity $O(m \log(n_b))$ where m is the number of union operations, i.e. the number of pairs of maximal black nodes at a unit distance from each other. In the worst-case, this complexity could become $O(n_b^2 \cdot \log(n_b))$.

**6. Component clustering:** This step involves running the traditional MST algorithm on the component graph. The time complexity to obtain the cluster tree is $O(n_c^2 \cdot \log(n_c))$. To extract $(k)$ disjoint clusters, the $(k-1)$ longest edges are removed. Clusters can be obtained by running any graph traversal algorithm (BFS/DFS) on the disjoint cluster tree.

The set color and find maximal black node are grid-based procedures. Like all such algorithms, their time complexities are dependent on the grid size and hence on the spatial extent of the data, and independent of the number of points in the tree. However, the additional operations of computing pairwise distances between visible boundaries, and boundary construction are $O(SA^2)$. In practice, since many mesh cells combine to form components (instead of being independent, isolated mesh cells), this yields a significant empirical speed up.

The **overall time complexity** is the dominant value among the complexities of various procedures above, namely among the terms $O(n \cdot \log E)O(N_{nw})$, $O(N_{nw})$, $O(SA^2)$, $O(SA \cdot \log(SA))$, $O(n_b^2)$, $O(m \log(n_b))$ and $O(n_c^2 \cdot \log(n_c))$. The exact complexity achieved will depend on the data at hand. The worst-case time complexity occurs in the rare case when most mesh cells occur in a very low-dimensional subspace (extreme case: points lined up along an axis), forming a very thin components with mesh predominantly lying at the component boundaries and few in the interior. This takes away the advantage of the reducing the computation due to intercomponent distances being determined by the (usually smaller number) of boundary cells. The time complexity of the various pre-hierarchy (hierarchical clustering) steps is then dominated by the large value of $SA$ $(n_b)$ which approach $n$. In comparison, mesh based algorithms such as STING have a worst case time complexity of $O(n)$, density based algorithms such as DBSCAN have a worst case time complexity of $O(n^2)$ and graph based techniques such as spectral clustering have a worst case time complexity of $O(n^3)$.

## 6 EXPERIMENTS

Our experiments are aimed at verify the properties of DAC. (1) In Sec 6.1, we first evaluate the correctness of DAC clusters in the Fundamental Clustering Problems Suite (FCPS) which are easy to evaluate visually due to FCPS dimensionality. (2) We next evaluate DAC for expected run time speedups vs one representative algorithm from each of the three categories reviewed in Sec 2 on the FCPS datasets for increasing number of $n$ with fixed $D$ (3) We then evaluate the quality of DAC clusters in high dimensions by matching its clusters with those detected by a CNN. (4) Finally, we evaluate DAC's run time in high dimensions for an increasing number of $n$, as well as for various combinations of $n$ and $D$.

### 6.1 EXPERIMENTS WITH FCPS DATASETS

We have experimented with four datasets. The challenges offered by these datasets and our clustering results on them are presented below.

#### 6.1.1 HEPTA DATASET

This dataset includes clusters with different densities. To ensure separability among different datapoints, the number of grid points must be above the number of datapoints. We scale up the grid to a 50x50x50 cube (of 125000 grid points), with a vertex at the origin. Fig 2 shows the clusters obtained by DAC; in addition, it illustrates the operation of DAC by showing outputs at two intermediate stages of the algorithm.

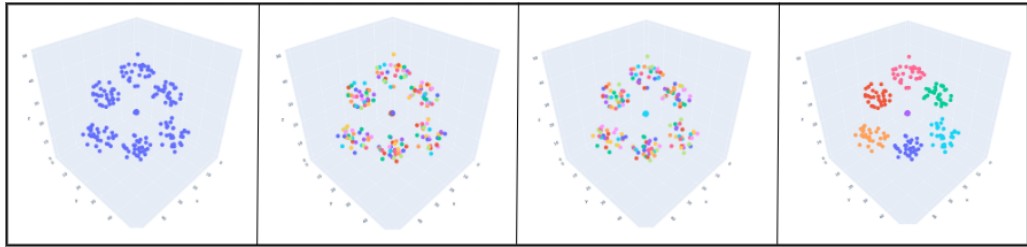

Figure 2: Hepta Dataset: Left to Right - (1) Inputs points, (2) Maximal black node extracted by DAC. Points of the same color belong to one maximal black node. (3) Components extracted by DAC, points in each component shown in the same color. Note that the set of one-step reachable maximal black nodes are merged into a larger single component, indicated by more mesh cells of the same color. (4) Final seven ($k = 7$) clusters.

#### 6.1.2 CHAINLINK DATASET

The chainlink dataset consists of two intertwined tori that are not linearly separable. Like any suitably designed proximity based algorithm, DAC is able to segregate the two tori clusters (Fig. 3).

#### 6.1.3 TARGET DATASET

The target dataset contains two large clusters with 4 small ones as outliers. Note that DAC can detect the four outlier clusters (Fig. 3).

#### 6.1.4 ENGYTIME DATASET

The Engytime dataset is a mix of two Gaussian clusters, one isotropic with unit variance, and the other, inclined one, has variances 2 and 1.6 along its principal axes.

### 6.2 COMPARISON OF RUN TIMES

To test the speedups expected from DAC, we compare runtimes of our algorithm for increasing number of $n$ with three other algorithms: DBSCAN (density based), spectral clustering (graph based),

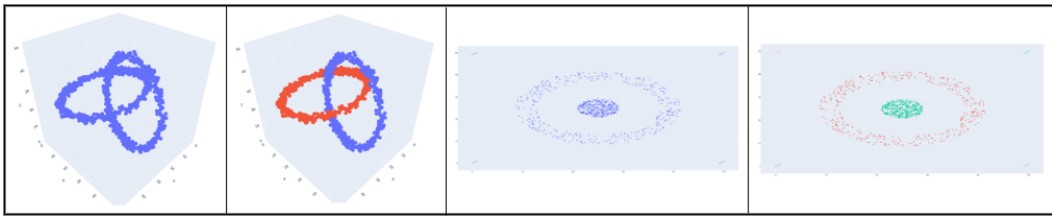

Figure 3: Input data (Fig 1 and 3 from left), and final clusters (Fig 2 and 4) produced for the chainlink and target datasets.

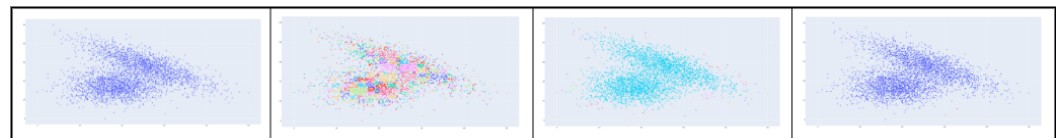

Figure 4: Clusters detected by DAC in the Engytime dataset. Left to Right: (1) Input data points, (2) Maximal black nodes, (3) Components, (4) Output clusters for $k = 2$
Note that separation between the two clusters at their closest approach is comparable to their local, within-cluster nearest-neighbor distances; hence they are detected as a single cluster, not two as (as per our cluster definition in Sec 1).

and CLIQUE (mesh based). Since we do not have datasets with varying number of datapoints for the same cluster geometry, we modify the FCPS datasets by increasing the density of each of the datasets to obtain three versions having $n = 5,000, 50,000, 500,000$. We run all four algorithms and record their runtimes. Fig 5 shows how the runtime of each algorithm increases with $n$. We see that DAC has the slowest increase.

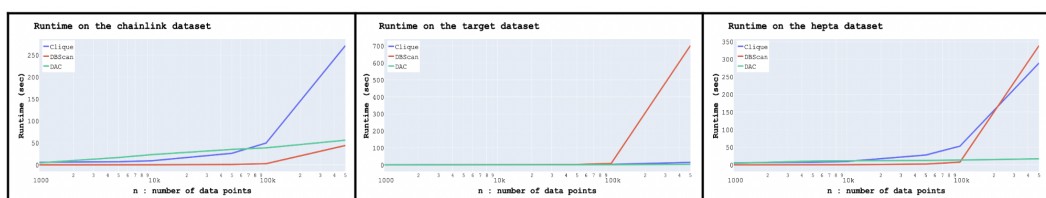

Figure 5: Runtimes of four different types of clustering algorithms on the FCPS datasets modified to obtain three different values of $n$: 5,000, 50,000 and 500,000, by increasing cluster density while maintaining cluster geometry. The plots show runtimes for the algorithms CLIQUE(blue), DB-SCAN(red) and DAC(green) for different $n$ (shown on log scale). In general, DAC has the slowest rate of increase and its runtime advantage over other algorithms increases with $n$. The runtime for spectral clustering is too large to fit in the figure.

## 6.3 Cluster Quality and Run time Performance in High Dimensions

**Quality:** We also evaluate DAC by matching them with the clusters detected by an independent algorithm - a CNN classifier. The dataset we use is a set F of 10,000 features (datapoints) in a 512-dimensional space. F is output of the convolutional layers of a VGG-16 CNN classifier, with input being the 10-class CIFAR-10 test dataset. The output can be modeled as containing a cluster for each of the 10 classes. The classifier has been reported to yield a test accuracy $A$ of $93.54\%$ We test our algorithm by obtaining the best clusters in F, and then computing the average label-purity P of the clusters. The P value we have obtained using DAC is $95.79\%$ which is almost the same as $A$. The reason $P$ is better than $A$ is likely due to the fact that we are detecting clusters in the test data, not predicting them from training data, like CNN classifier has to), This agreement validates the quality of DAC clusters because they match the clusters obtained by CNN which indeed correspond to class clusters (Fig 6 (b).

**Run Time:** Finally, we compare the run times for the same four algorithms for the same CNN data for increasing $n$ ((Fig 6 (c)), and for different combinations of $n$ and $D$, for a synthetic dataset consisting of a cluster in a hypercube, surrounded by another cluster in a hypercubical ring (Fig 6 (d)). The maximum values of $n$ and $D$ we choose are determined by the CPU memory limits.

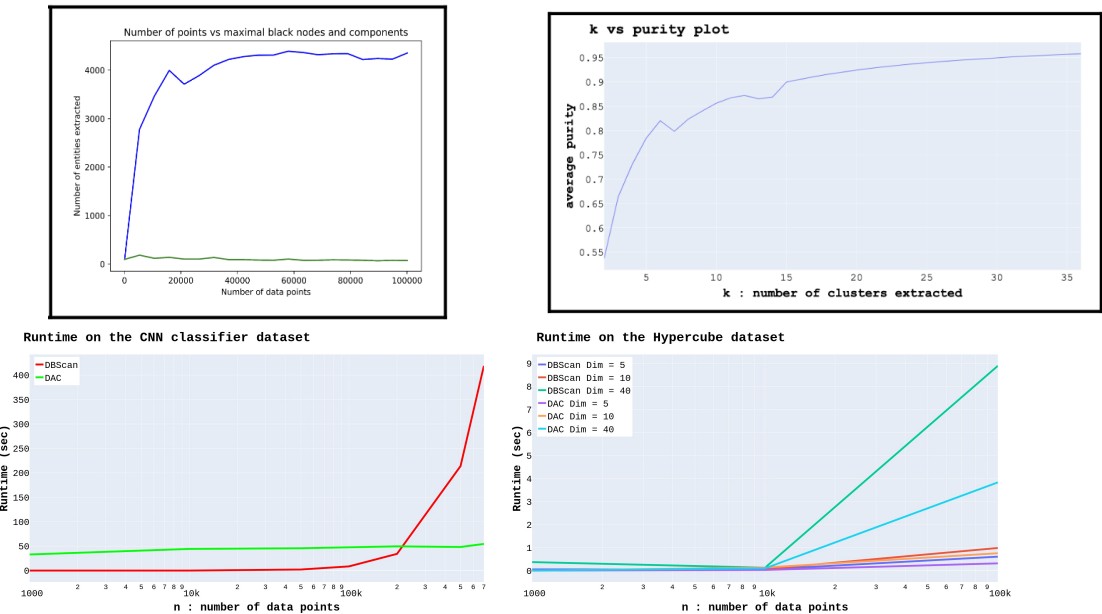

Figure 6: (a) DAC's low time complexity results in part from the expectation that the number of components increase slower than number of points, as seen in the observed increases here. (b) Validation of the correctness of clusters detected by DAC in high-dimensional data. The best clusters (35 here) detected by DAC in a 512-dimensional space for features output by a CNN for CIFAR 10 dataset. This achieves a 0.95 purity of labels (labels unknown to DAC but known to CNN) which is also the accuracy obtained by the CNN. This suggests that the clusters detected by DAC and CNN are similar. (c) For the dataset in (c), observed run times of different algorithms for increasing $n$. The runtime for Spectral clustering is too large to fit in the figure. The runtime and performance of the Clique algorithm is highly dependent on it's parameters, and its runtime is too large to fit in the figure when its parameters are chosen to get reasonable clusters. (d) Run times observed for a synthetic dataset of a hypercubical ring cluster surrounding a hyeprcube core cluster, for varying number of $n$ and $d$ (here we can choose both freely). Again, Spectral clustering and Clique run times are out of range as in (c). In both (c) and (d), DAC shows the slowest rate of increase with $n$ for all values of $D$.

### 6.4 Number of Components Extracted by DAC vs Number of Points

Towards verifying that the number of components is indeed much smaller than the number of points (which translates into a runtime advantage for DAC, as discussed in Algorithm and Complexity sections), we show in Fig 6(a) that the observed numbers of maximal black nodes as well as components in the modified chainlink dataset indeed grows very sublinearly with $n$.

## 7 Limitations and Conclusions

We have presented an algorithm for local to global detection of clusters. Its advantage over many other clustering algorithms becomes particularly pronounced as $n$ increases. The complexity results we have presented are for typical (average) cases; as we have noted in the main paper and more in Appendix, the worst case, degenerate data, although rare, may wipe out DAC's advantages. Further, since there is not much large-$D$ data with ground truth available, evaluation of DAC on large $D$ data has been difficult. We have used CNN classifier output for this purpose, treating CNN's class label outputs as ground truth. This is incorrect but may be adequate if the classifier accuracy is high. We plan to further test the performance of DAC by integrating it incorporating it into complex learning algorithms requiring clustering.

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

# A   APPENDIX

## A.1   SET COLOR

```
Set color
Input : root of the tree
   1.   if (edge length of root = =1):
           root.color = 1.                    // Black nodes are numbered 1.
        else :
           num_of_black_children = 0
           for all children (cl) of the root :
              num_of_black_children += Set color(cl).
           if (num_of_black_children == 2^D):
              root.color = 1          // all children are black
           else root.color = 0
   2.   return root.color
```

## A.2   FIND MAXIMAL BLACK NODES

```
Find Maximal Black Nodes
Input : root of the tree
Output : Maximal black nodes γ_1, γ_2, ···, γ_{n_b}, (n_b) denotes the number of maximal black nodes

Initialization : node id = 1
   1.   if(root.color == 1) :
           set_node_id(root, node id)     //recursively sets the node_id for all nodes rooted at root
           γ_{node id} = root
           root.isblack = True
           node id = node id + 1
   2.   else :
           for all children (cl) of the root :
              Find Maximal Black Nodes(cl)
```

### A.3    FILL MATRIX

Representation of node boundaries : In D-dimensional space a node has 2D boundaries, two in every dimension. We use a dictionary to maintain the node boundary wherein the boundary corresponding to the $i$-th dimension is stored with the key $(2i)$ and $(2i + 1)$ (similar to how heaps are represented using linear arrays). Hence the node boundary dictionary contains the keys $0, 1, 2, \cdots 2D-2, 2D-1$.

---

**Fill_matrix**
Outline : A top down procedure to compute node boundaries and fill the adjacency matrix *A*. Recursively computes and uses the input node boundary to update the matrix *A*.

**Input** : root of the tree (*root*)
**Output** : Boundary of the input node (*root*), updated matrix *A*.

**Initialization** : $A \in R^{n_b \times n_b}$ ($n_b$ denotes the number of maximal black nodes). All entries are initialised to $+ \infty$

1. *Base Case* : When the root is a black node, return the root boundary.
   If $root.isblack$:
     return get_node_boundary(root, $d$ = None)

2. Find the boundary of all child nodes to construct the boundary of the node *root*.
   boundaries = ϕ    // empty dictionary
   for rank, child in root.children :
     boundaries[rank] = fill_matrix(child)

3. Compute the visible boundary between every pair of children and use it to update the matrix *A*.
   for rank1 in root.children.keys():
     boundary1 = boundaries[rank1]    // fetch child boundary.
     location1 = binary_representation(rank1)

     for rank2 in root.children.keys():
       boundary2 = boundaries[rank2]
       location2 = binary_representation(rank2)
       relative_position = location2 - location1
       visible_1, visible_2 = []    // contains nodes on the visible boundary of child with rank1 and rank2.

       for dimension, direction in enumerate(relative_position):
         if(direction == 1):
           visible_1.append(boundary1 [2*dimension + 1 ])
           visible_2.append(boundary2 [2*dimension])
         if(direction == -1):
           visible_1.append(boundary1 [2*dimension ])
           visible_2.append(boundary2 [2*dimension + 1 ])

       Update matrix using boundaries visible_1 and visible_2.
4. Construct *root* boundary using child node boundaries.
5. return *root* boundary.

---

The *fill matrix* procedure contains the following steps:

1. Recursively compute the node boundaries of all children
2. Use the child boundaries to update the adjacency matrix with the minimum distance between maximal black nodes :
   Worst-case time complexity : Worst case time complexity arises when the visible boundary is computed between each pair of siblings at each level in the hypertree. At a certain level in the tree, each pair of siblings (with edge length say $\left(\frac{s}{2}\right)$) can share at most (D) faces, each containing at most $\left(\frac{s}{2}\right)^{D-1}$ mesh cells. Hence, for this visible boundary we compute distance between $(D \cdot \left(\frac{s}{2}\right)^{D-1})$ pairs of mesh cells. There are $\binom{2^D}{2}$ such pairs of siblings inside the parent node. This gives a time complexity of $O(\binom{2^D}{2} \cdot \left(D(\frac{s}{2})^{D-1}\right)^2) = O(D^2 s^{2(D-1)})$ for all sibling pairs in one parent. However multiple such parent exist at each level in the tree. The parent with side length $(s)$ belongs to the level $L = log_2\left(\frac{E}{s}\right)$, where $(E)$ denotes the side length of the root. The level $(L)$ contains $2^{LD}$ nodes, each with edge length $(s)$. Hence the time complexity at each level in the tree is given by $O(2^{LD} \cdot D^2 s^{2(D-1)}) = O(D^2 E^D s^{D-2})$. Summing over all levels $s = E, \frac{E}{2}, \cdots, 1$ gives the total time complexity of this step as

$$TC = O\left(D^2 E^D \cdot \left(E^{D-2} + \left(\frac{E}{2}\right)^{D-2} + \cdots\right)\right)$$

$$= O\left(D^2 E^D \cdot E^{D-2}(1 + 1 \cdots)\right) \tag{1}$$

$$= O(D^2 E^{2(D-1)} log(E))$$

Note that the time complexity grows as the square of the surface area of the hypertree.

3. Update the adjacency matrix using the visible boundary between a pair of sibling nodes :

   Once the visible boundary between a pair of sibling nodes is computed in the *fill matrix* procedure, the pairwise distance between every mesh cell on the boundary is used to update the matrix A. The algorithm uses the maximal black node identifier set during the *find maximal black nodes* procedure to find which entry of the matrix is to be updated.

---

**Procedure to update adjacency matrix *A* given a pair of visible boundaries**

**Input** : Visible boundaries $B_i$, $B_j$, adjacency matrix *A* .

**Output** : Updated adjacency matrix *A* .

1. For cell_1 in $B_i$ :
      For cell_2 in $B_j$ :
       mb_node_1,mb_node_2= cell_1.node id, cell_2.node id  // maximal black node identifier
       distance  <- euclidean distance(cell_1, cell_2)
       If distance <  A[mb_node_1][mb_node_2] :
        A[mb_node_1][mb_node_2]= distance   // update matrix entries
        A[mb_node_2][mb_node_1]= distance

---

4. Construct the root boundary using the child node boundaries :

   This procedure computes the boundary of non-black nodes given the boundary of all children nodes.

   For every child, the while(dimension < D) loop is used to iterate over all the neighbouring siblings which may contain points to be included on the boundary. For each child $(rank\_1)$ and dimension $(d)$ the procedure contains three steps :

   (a) Insertion of points from the current boundary to the parent boundary :  The entry to be updated in the parent's boundary is determined via the child's position inside the parent.
   Worst case time complexity :  For a parent with edge length $(s)$, each face may contain a maximum of $(s^{D-1})$ mesh cells. Hence, the time complexity of adding points from the children boundaries would be $O(s^{D-1})$ per face, giving a time complexity of $O(D \cdot s^{D-1})$ for each parent with length $(s)$.

   (b) Inserting exposed points from the neighbouring siblings :  Let us indicate the boundaries of the current child $(rank\_1)$ and the sibling $(rank\_2)$ as $(B_1)$ and $(B_2)$ respectively. To determine whether some mesh cell $n$ from $B_2$ is visible through holes in $B_1$ we must determine if there exists a mesh cell $n_*$ in $B_1$ which has the same coordinates as $n$ in dimensions other than $(d)$. We accomplish this by creating a scalar encoding $E(.)$ for each point on $B_1$ and $B_2$. The encoding is a function of the mesh cell, parent edge length $s$ and the dimension parameter $(d)$.
   To motivate the use of the encoding function we propose the following lemma.
   *Lemma : Any mesh cell $n \in B_2$ would not be exposed through holes in $B_1$ iff there exists a mesh cell $n_* \in B_1$ such that $E(n_*) = E(n)$*
   *Proof :*  Let the child $(rank\_1)$ be at position $(l_0, l_1, \cdots, l_{D-1})$ and have edge length $\frac{s}{2}$. As the child with rank $(rank\_2)$ is the neighbour along the dimension $(d)$ it has the

**Construct node boundary using child node boundaries**

**Input** : node *root, boundaries of all children of *root : boundaries*
**Output** : Boundary of the node *root* : boundary
**Initialization** : boundary = φ
1. for rank_1 in boundaries.keys():        # iterate over all non-white children
      boundary_1 = boundaries[rank_1]

      binary_rank = binary_representation(rank_1)
      dimension = 0
      transform = 1   // used to access neighbours of child with rank_1 using the xor operation.

      while (dimension < *D*) :
        boundary_key = 2*dimension + binary_rank[dimension]     // which entry of the boundary is to be filled.
        boundary[boundary_key] <- boundary[boundary_key] ∪ boundary_1[boundary_key] // extend boundary
        rank_2 = rank_1 ^ transform     //toggles the bit at position (dimension) to give the neighbouring child

       if child with rank_2 exists :
        boundary_2 = boundaries[rank_2]
        encoded_boundary = encode_boundary (boundary_1[boundary_key], dimension)

        for point in boundary_2[boundary_key]:
          encoded_point = encode_point(point, dimension)
          if encoded_point not in encoded_boundary:
            boundary[boundary_key] = boundary[boundary_key] ∪ [point]
            encoded_boundary.insert(encoded_point)

      else :
        binary_rank_sibling= binary_representation(rank_2)   # binary rank of the empty sibling (rank2)
        boundary_key = 2*dimension + binary_rank_sibling[dimension]
        encoded_boundary = φ
        for point in boundary_1:
          encoded_point = encode_point(point, dimension)
          if encoded_point not in encoded_boundary:
            boundary[boundary_key] = boundary[boundary_key] ∪ [point]
            encoded_boundary.insert(encoded_point)

       dimension +=1
       transform *= 2    // the next power of 2 would produce the next neighbour of child (rank_1)

   return boundary

---

**encode_point**

**Input** : point $p = (p_0, p_1, \cdots, p_{D-1}) \in R^D$, dimension (*d*).
**Output** : The point encoding.
1. e <- edge length of caller node
2. encoding <- $p_o + p_1 e + p_2 e^2 + \cdots p_{d-1} e^{d-1} + p_{d+1} e^d + \cdots + p_{D-1} e^{D-2}$
3. return encoding

position $(l_0, l_1, \cdots l_{d-1}, l_d \pm \frac{s}{2}, l_{d+1}, \cdots l_{D-1})$. Let the mesh cell $n_*$ be at the coordinate $(l_0 + p_0, l_1 + p_1, \cdots, l_{D-1} + p_{D-1})$ and the mesh cell $(n)$ be at the coordinate $(l_0 + i_0, l_1 + i_1, \cdots l_{d-1} + i_{d-1}, l_d + -\frac{s}{2} + i_d, l_{d+1} + i_{d+1}, \cdots l_{D-1} + i_{D-1})$. The difference between the encoding of the mesh cell follows from the definition of the encoding function.

$$
\begin{aligned}
E(n_*) - E(n) = & (p_0 - i_0) + s \cdot (p_1 - i_1) + \cdots s^{d-1} \cdot (p_{d-1} - i_{d-1}) \\
& + s^{d+1} \cdot (p_{d+1} - i_{d+1}) + \cdots s^{D-1}(p_{D-1} - i_{D-1}))
\end{aligned}
\tag{2}
$$

Note that each $(p_k, i_k)$ denotes the offset of the mesh cells from the node position and are whole numbers in the range $[0, s/2]$. Hence the encoding function can be interpreted as a base-$s$ representation of the mesh cell offsets, i.e. $(p_0, p_1, \cdots p_{D-1})_s$. The two encodings would be equal iff for all $(k \neq d)$ $p_k = i_k$, i.e. when the mesh cell $(n_*)$ has the same coordinates as $(n)$ in all dimensions other than $(d)$. This ensures that the mesh cell $(n)$ is not exposed to the visible boundary through holes in boundary $(B_1)$.

The lemma shows that if we encode each mesh cell on $(B_1)$ beforehand we can determine if any mesh cell on $(B_2)$ must be included in the parent boundary by a simple search in the set of encodings of all mesh cells on $(B_1)$ (denoted by $E(B_1)$). Note that performing a coordinate comparison to determine which mesh cell goes into the parent boundary would require an $O(D)$ computation per pair of mesh cells on $(B_1), (B_2)$. This would give a time complexity of $O(|B_1| \cdot |B_2| \cdot D)$, where $|B_1|$ denotes the size of boundary $B_1$. However, if we use the encoding function we would only require $O((|B_1| + |B_2|) \cdot D)$ time to encode both the boundaries and $O(|B_2| \log(|B_1|))$ time to search for a matching encoding in the set $E(B_1)$. Hence, the time complexity for this computation would scale as $O(SA \log(SA))$ rather than $O(SA^2)$, where $SA$ denotes the surface area of the child nodes.

Worst case time complexity : At a certain level in the tree, for each child (rank_1) and dimension $(d)$ to determine mesh cells exposed through holes we must compute mesh cell encodings on the boundary face along dimension $(d)$. A face of the boundaries $(B_1), (B_2)$ can contain a maximum of $\left(\frac{s}{2}\right)^{D-1}$ mesh cells. Encoding each mesh cell requires $O(D)$ time, giving a time complexity of $O(D \cdot \left(\frac{s}{2}\right)^{D-1})$. Once all mesh cells on $(B_1)$ have been encoded we search for encoded mesh cells $E(B_2)$ in the set $E(B_1)$, resulting in a time complexity of $O(\left(\frac{s}{2}\right)^{D-1} \log\left(\left(\frac{s}{2}\right)^{D-1}\right)) = O(D \left(\frac{s}{2}\right)^{D-1} \log(s))$. Accumulating over all (rank_1, $d$) pairs gives the time complexity of $O(D \cdot 2^D \cdot D \left(\frac{s}{2}\right)^{D-1} \log(s)) = O(D^2 s^{D-1} \log(s))$ for each node with side length $(s)$ in the tree. The node with side length $(s)$ belongs to the level $L = log_2\left(\frac{E}{s}\right)$, where $(E)$ denotes the side length of the root. The level $(L)$ contains $2^{LD}$ nodes, each with edge length $(s)$. Hence the time complexity at each level in the tree is given by $O(2^{L \cdot D} \cdot D^2 s^{D-1} \log(s)) = O(D^2 \cdot E^D \cdot \frac{\log(s)}{s})$. Summing over time complexity at each level in the tree, i.e. $s = E, \frac{E}{2}, \cdots$, gives the cumulative time complexity as

$$
\begin{aligned}
TC &= O\left(D^2 \cdot E^D \left(\frac{\log(E)}{E} + \frac{\log(E/2)}{E/2} + \cdots\right)\right) \\
&= O(D^2 \cdot E^D \cdot \log(E)(1 + 1 + \cdots)) \\
&= O(D^2 E^D \log^2(E))
\end{aligned}
\tag{3}
$$

## A.4  GET NODE BOUNDARY

As the base case of the recursive *fill matrix* procedure, the *get node boundary* procedure computes the boundary of a black node by merging the boundaries of its child nodes. This procedure differs from part 3 of fill matrix as it computes the boundary of a black node, which is shaped as a hypercube. The procedure is described below.

**get_node_boundary**
Outline : find the boundary of a black node in a certain direction.

**Input** : Black node $\gamma_i$ , boundary direction $(d)$

**Output** : Node boundary $B_i$

**Initialization** : $B_i = \phi$

1. if sum $(d) == - D$ :  // each entry in $(d)$ is -1
   return $\phi$
2. *Base case* : Boundary of a mesh cell.
   If edge length of $\gamma_i == 1$ :

   if $(d)$ is None :
   return $B_i = \{key : \gamma_i \mid key \in \{0, 1, 2, \cdots, 2D - 1\}\}$ // boundary in every direction is $\gamma_i$
   else :
   for dimension, direction in enumerate$(d)$ :
   If direction != -1 :
   $B_i[2 \cdot dimension + direction] = \gamma_i$
   return $B_i$

3. for rank, node in $\gamma_i.children.items()$ :

   direction = binary_representation(rank)
   if $(d)$ is not None :
   $direction[\ direction\ != d] \leftarrow (- 1)$ // assign -1 to directions where no node boundary is required.
   node_boundary = get_node_boundary(node, direction)  // get boundary of child node in the specified direction via a recursive call.
   for key, item in node_boundary.items() : // expand node boundary using child's node boundary.
   $B_i[key]$.extend(item)
4. return $B_i$

To obtain the boundary of a node, we compute the boundary of each child node using a recursive call but with a $(d)$ parameter corresponding to its rank inside the parent. The function is called with the parameter boundary direction $(d)$ which is a D-dimensional vector whose entries indicate which boundary is to be fetched. For a certain dimension, a corresponding (0) entry in $(d)$ would fetch the left boundary, a (1) entry would fetch the right boundary and (-1) signifies that no boundary is to be returned in this dimension.

As an example consider the following two-dimensional node with edge length = 2 called with $(d)$ = [0,1], which means that the left and upper boundary is required along the x and y-dimension respectively. For this purpose we only need the left boundary from the child with rank [0,0] (the bottom-left child), the top-left boundary from the child with rank [0,1] (the top-left child) and the top boundary from the child with rank [1,1] (the top-right child). This is accomplished by matching the child rank and (d). The top-left child is called with boundary direction = [0,-1], which signifies that only the left boundary is needed along the x-dimension. Others child nodes are called in a similar manner.

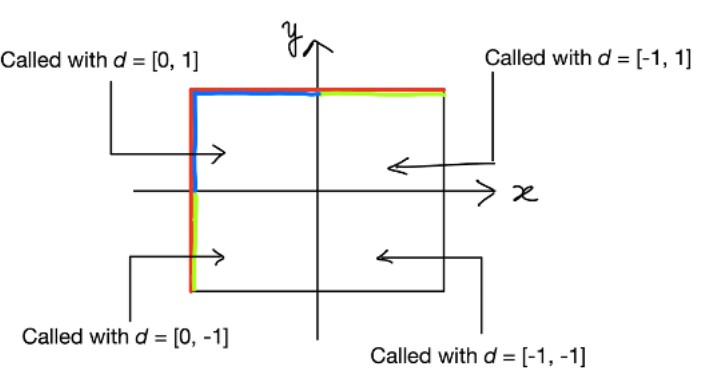

