# OpenReview forum: "Divide-and-Cluster: Spatial Decomposition Based Hierarchical Clustering"
_ICLR.cc/2023/Conference — Submitted to ICLR 2023_

### Official Review · Reviewer_1fFd · 2022-10-24

**Confidence:** 3
**Correctness:** 3
**Technical Novelty And Significance:** 2
**Empirical Novelty And Significance:** 2
**Recommendation:** 3

**Clarity, Quality, Novelty And Reproducibility:**

*Clarity*: the description of the method in the paper is relatively easy to follow though it requires some background knowledge on tree traverse etc.

*Quality*: the paper needs to be carefully proofread. There are many typos in the paper. Moreover, to illustrate the effectiveness of the proposed methods, the author also need to compare with other clustering approaches, current experiment is not enough.

*Novelty*: the proposed method is novel to the best of the reviewer's knowledge.

*Reproducibility*: the algorithm details are given in the appendix, but the description of the experiment details  in section 6.3 are not enough to reproduce the experiment.


**Strength And Weaknesses:**

I really like the paper is not overselling the proposed method, the limitations of the method is discussed in the paper.

Weaknesses:

1. Computational efficiency is one of the selling point of the proposed method. However, both in the description and the experiment, the reviewer do not see enough evidence that the proposed method is more efficient especially in higher dimensions. The reviewer suggests the authors either put a table to list the computational cost of various approach and compare the computational complexity theoretically or conduct more experiments to show this empirically.

2. Additional work on related work. There are other tree-based clustering methods that the authors do not discuss, such as [this](http://web.cs.ucla.edu/~wwc/course/cs245a/CLTrees.pdf). Anothe paper that has a similar flavor is [this](https://arxiv.org/pdf/1802.04397.pdf). The reviewer suggests the authors to have a more detailed disucssion on the related work. Although the distance is not direclty used, the ajacency matrix implicitly uses the information of the distances, can the authors highlight the difference of the proposed method and other distance based approaches?

3. In the experiment, only the computational time are compared quantitatively, it would be very helpful and more convincing if quantitative metrics for the agreement of different clustering approaches is also used. Section 6.3 only reports the purity score, more metrics should be used for comparison.

4. The description in the paper requires the readers to have a lot of background knowledge in tree-based algorithm. It would be really helpful if either 1) visualize the description in section 3 or 2) refer the readers to some useful resources.


**Summary Of The Paper:**

This paper proposes an algorithm to use tree-based algorithm for the purpose of clustering. The tree-based approach first partite the data into different chuncks hierarchically and merges smaller pieces into clusters via efficient algorithms. Compare to conventional clustering algorithms, the authors claim that the method is computationally efficient.

**Summary Of The Review:**

The major concern is that the experimental results and the description are not convincing for the reviewer to take the proposed method if there is a need for clustering. More thorough discussion and comparison with existing methods to demonstrate the method is indeed better than existing ones. Hence, the reviewer believes additional work needs to be done before the paper can be published.

---

> ### Author Response · Authors · 2022-11-19
> **Response to R4**
>
> 1. Computational Efficiency: We have presented the complexity of DAC theoretically and empirically. We have added new experiments to provide further and diverse support.
>
> 2. Related Work: The paper “Clustering Via Decision Tree Construction” by Bing Liu1 , Yiyuan Xia2 , and Philip S. Yu3 (CLTrees.pdf (ucla.edu)) provides an algorithm for the problem of partitioning a labeled point set into hyperrectangular subsets such that each subset maximizes purity of labels in it (one label is the best); it is not concerned with mutual distances between adjacent hyperrectangles. Our problem, on the other hand, is to partition a point set into subsets such that the closest distance between points across two adjacent subsets is larger than the local within subset distances, and to recursively merge the clusters to form larger clusters with increasing intercluster distance.
>
> Regarding the paper 4.2.1 1802.04397v3.pdf (arxiv.org), our goal is to use adjacency matrix of components instead of points, which introduces significant efficiency.
>
> Regarding other distance based methods, DAC does not use only distances; instead it makes use of the full geometry of the D-space through the tessellation. We believe this helps DAC achieve its better performance.
>
> 3.Quantitative Comparison with other Methods: We have added new experiments to contrast DAC performance with those of other algorithms.
>
> 4. Purity: We do not have to consider purity for comparison since we are concerned with point clustering problem and not separation of the point set into uni-labeled clusters. The reference to purity in Sec 6 is about the ancillary task of finding a ground truth for high-dimensional CNN clusters (since we do not have such datasets), so that we can evaluate our algorithm in high dimensions.
>
> 5. We have added an overview figure (Fig 1) to help visualize the steps in DAC through an image.
>
> 6. We have tried to edit for typos.
>
> 7. We have included detailed procedures and pseudocode in Appendix.
>
> May we please request R4 to take a look at our responses to R1-R3, as several responses there may complement our response here and are not repeated here.

---

> > ### Comment · Reviewer_1fFd · 2022-12-07
> > **post-rebuttal**
> >
> > Thanks for the response, I appreciate the authors' effort in revising the paper. After the rebuttal, I still think a more thorough comparison with the exisiting approaches is needed. Hence, unfortunately, I don't think the work is ready to be published at the current stage.

---

### Official Review · Reviewer_Sm2r · 2022-10-24

**Confidence:** 2
**Correctness:** 2
**Technical Novelty And Significance:** 2
**Empirical Novelty And Significance:** 1
**Recommendation:** 3

**Clarity, Quality, Novelty And Reproducibility:**

I think all of the above can be improved. For example, my recommendation would be to write the algorithm in Section 4 as pseudocode for clarity, and also to use consistent terminology in the calculation of the time complexity. Since the references are rather old, and recent work has not been cited, this raises a question mark on the novelty of the algorithm as well as its effectiveness in discovering useful clusters.

**Strength And Weaknesses:**

+ The authors have procedures used in their algorithm in detail
- It is not clear whether the quality of clusters or the efficiency of the approach is better than simple hierarchical algorithms that are capable of using the numeric features representing spatial location and have lower time complexity. The paper discusses that STING and DBscan also have better time complexity. It would be good if the authors clarify this
- The papers discussed in the related work are old. It would be useful if recent work is cited, since more recent work is available on spatial hierarchical clustering

**Summary Of The Paper:**

The paper presents a hierarchical algorithm based on the locality of data points. Entities are represented in the D dimensional space as D-dimensional hypercubes and are assigned colors based on whether they are completely/partially contained within another entity. The algorithm first inserts entities into a tree, colors nodes, computes the adjacency matrix and components based on maximal black nodes, and finally runs the MST algorithm on the components. The worst time complexity is O(n^2log(n)).

**Summary Of The Review:**

The main claim of the paper is that efficiency has been increased. However, since
- recent work is not cited,
- the approach is rather complicated, and
- it's not clear in which situations/domain such an approach would result in more meaningful clusters
- the detected clusters are not evaluated by comparison with other techniques (recent or old)
I don't really know whether the claim is supported.

Some other points in the paper also raise questions. For example, according to the authors 'degenerate data may wipe out DAC's advantages'. Moreover, the authors have pointed out themselves that STING and DBScan has lower time complexity. So unless there is a clear advantage in using the presented approach, it is difficult to understand the claim.

---

> ### Author Response · Authors · 2022-11-19
> **Response to R3**
>
> 1. Efficiency Compared to Other Algorithms: For the differences in our algorithm that make it unique and lead to higher efficiency, may we request R3 to please take a look at our responses to R1 and R2. We explain there how we validate the correctness of DAC clusters, why DAC is more efficient (due to the use of tessellation, cluster component processing, use of cluster geometry), what DAC's theoretical efficiency is, how we test DAC experimentally, and how the results validate DAC's superiority. We welcome pointers to other algorithms with comparable performance for arbitrary dimensional and arbitrarily large number of points.
>
> 2. Recent Papers: We will add references to recent survey papers
>
> 3. Complexity: We have updated the section on the time and storage complexity of DAC.
>
> 4. Pseudocode: We have provided Algorithm in pseudocode in Appendix.
>
> 5. Novelty, Effectiveness and meaningfulness of DAC clusters: May we request R3 to please see our response to R1 and R2 for the novelty and Effectiveness of DAC.
>
> 6.  Complicated: We agree that the geometrical parts of DAC are somewhat detailed and hard to follow. We have tried to improve the explanations and added a new figure to provide a high level explanation of DAC steps through an image (rather than point patterns).
>
> 7. Comparison with other algorithms: We have chosen one algorithm for comparison from each of the three other major classes of algorithms we have reviewed in the Related Work section.
>
> Again, we will appreciate it if R3 could also look at our responses to R1 and R2.

---

### Official Review · Reviewer_snwo · 2022-10-25

**Confidence:** 3
**Correctness:** 2
**Technical Novelty And Significance:** 2
**Empirical Novelty And Significance:** 2
**Recommendation:** 3

**Clarity, Quality, Novelty And Reproducibility:**

I think that the following statement is quite strong and not always true. It would be helpful to add citations to the cost functions the authors are referring to here.

> Hierarchical clustering approaches derive a tree (hierarchy) of clusters in which clusters at each
level correspond to a local minimum of the cost function with a level-specific value. Each cluster
at a level splits into multiple sub-clusters at the level immediately below, where the sub-clusters are
more compact than the parent clusters, corresponding to stronger local minima of the objective function. Hierarchical clustering can be top-down (recursively split clusters) or bottom-up (recursively
combine clusters).

Nit: When citing multiple papers it is nice to combine the citations e.g., \cite{A,B,C} rather than \cite{A}, \cite{B}, \cite{C}

The following statement is a bit strong, there are things that could be done for metric MST and various very scalable and parallelizable methods like Boruvka's algorithm as well as ways to use nearest-neighbor indexes (e.g. cover tree) to make computation more efficient (e.g., [Curtin et al, (2013)'s dual tree methods](http://proceedings.mlr.press/v28/curtin13.pdf)).

> unlike the MST algorithm which computes pairwise distance over all pairs of points

**Strength And Weaknesses:**

**Strengths**
* This paper explores interesting ideas about how methods for scaling hierarchical clustering by efficient data structures / algorithms for selecting which clusters can merge in a levelwise manner.
* The algorithm has novel characteristics and the authors have taken care to develop a method suited for the kinds of data they are considering.

**Weaknesses**
* It would be helpful to have more motivation for algorithms in this low-dimensional settings, when does such data occur in practice? What are the inductive biases of approaches that work well? Why does the proposed method achieve those inductive biases? It is hard to understand the generality of the method to datasets with high dimension?
* I think there needs to be a comparison to other well known clustering algorithms such as hierarchical agglomerative clustering. This would help readers understand how challenging the datasets can be. It would strengthen the paper further to consider a wider range of hierarchical clustering methods.
* It would help to understand to more clearly the properties of the algorithm in what it can recover and how quickly compared to competing methods. Theoretically, Big-O notation or more empirical performance. As it is, I am left as a reader uncertain about when to use the proposed method. Further I am left with not fully understanding what aspects of previous work the proposed method is offering an improvement on.
* I think that the paper would be higher impact in a conference that has greater focus on algorithmic techniques than what I think of ICLR's "typical paper", which is more aligned with representation learning. I think conferences such as KDD, CKIM, SDM, could be a better fit?


**Summary Of The Paper:**

This paper presents a spatial-partition-based hierarchical clustering approach. Efficiency comes from a decomposition of the space into bins and using the structure of the space to determine which clusters can be merged. The authors present experiments on spatial clustering datasets with clear visual clusters. The authors demonstrate scalability compared to DBSCAN, spectral clustering, and a mesh-based method as well as use of the method, DAC, in a downstream task.

**Summary Of The Review:**

The paper explores an interesting algorithmic idea for hierarchical clustering. However, the advantages of the proposed method are not greatly clear to me. The paper could be improved with a comparison to a wider array of techniques (including classic methods like HAC) and clarifying (1) importance of low-dimensional data (2) inductive biases of the algorithm that lead to its effectiveness.

---

> ### Author Response · Authors · 2022-11-19
> **Response to R2**
>
> 1. Motivation: Our goal is indeed clustering of high-dimensional data without excluding low dimensional cases. It will be good to know which part of the paper may suggest that. The only mention of low-dimensionality we have in the paper is in the abstract, namely “Many clustering algorithms are based on properties of relative locations of points, globally or locally, e.g., interpoint distances and nearest neighbor distances. This amounts to using a lower dimensional space than the full dimensionality D of the space in which the points are embedded.” But there we wish to point out the underutilization by existing algorithms of D-dimensional information in points vs our use of the full D-dimensional space (through hyeprcubical tessellation); our algorithm works on the D-dimensional patterns for arbitrarily high D. Also, we are interested in R2's comment on inductive bias since we are using recursion only for computational efficiency and making inferences.
>
> 2. Relationship to Other Algorithms: We have reviewed survey papers on clustering methods, as well as individual papers but have (to our surprise) not been able to locate any that make use of recursive division of the space and then merger of locally found clusters while taking advantage of the cluster geometry. There are algorithms that divide points into subsets but  their complexity is lower bounded by the efficiency of these subdivision algorithms which must find local groups of points (based on their coordinates) whereas the locations of points in the tree implicitly specifies coordinates of points contained therein. We use the same efficiency in the hierarchical aggregation part by keeping track of the evolving cluster geometry and computing intercluster distance in terms of points on their mutually visible boundaries. Other aspects of novelty may please be seen in Response to R1.
>
> 3. Properties: We have elaborated on the method, its theoretical (Big O) complexity, empirical performance on a variety of datasets, with varying values of D and n, and compared these with other methods. May we request R2 to see our response to R1, particularly Items 3 and 5.
>
> 4. Venue: Clustering being such a fundamental computational tool, we agree that the theme of this paper matches the conferences R2 points out. Since clustering is a common representation used in unsupervised learning as well as a frequent step in obtaining other representations, we believe clustering may also be a topic of interest to the ICLR community.
>
> 5. HC Cost Function: We refer to by cost function the criteria to determine best clusters among the continuum of clusters in a hierarchy.  A parent cluster is formed by merger of the closest child clusters.  The cost function we have used is the closest distance between clusters. Thus clusters higher up are sparser and bigger. This recursive, closest distance is the basis of the hierarchy. We have modified the statement to include these properties as a part of the statement.
>
> 6. Citations: We have followed the format suggested by R2.
>
> 7. We agree that this statement is very strong, and indeed an error. We agree that computation of distances between all pairs of points is not essential, and there are efficient MST algorithms which can be used. We have modified the statement accordingly and added related references. However, since our aim is clustering, and we need to find cluster pairs at increasing, closest, mutual distances, we need an MST for clusters instead of points. To compute that, we need to run an MST algorithm for points in each cluster pair at each level (to decide which ones to merge). That is less efficient than making it a part of the clustering algorithms by using cluster geometry and mutually visible cluster “surface” or boundary pairs.
>
> 8. For both “low dimensional data” and “inductive bias” aspects, please see above.

---

> > ### Comment · Reviewer_snwo · 2022-12-07
> > **Thanks for your response**
> >
> > Thank you very much for taking the time to respond to each of my questions. Apologies for the delay in replying here. Your clarifications are indeed helpful, however, I tend to maintain my original recommendation for the paper. I think you have interesting ideas here, but further revisions needed before publication.
> >
> > In response to your comments,
> > 1. Sorry, seems my mistake, the dimensionality in the experiments & use of axis-based partitioning lead me to think this.
> > 2. My comment was not about partitioning approaches, but a general comment about competing clustering methods. For what its worth, I think that KD-Tree-based algorithms like the ones I mentioned in the linked Dual-Tree paper are highly related as are methods linked by the other reviewers.
> > 3 & 6. Thank you for doing so.

---

### Official Review · Reviewer_ZhK4 · 2022-10-26

**Confidence:** 4
**Correctness:** 4
**Technical Novelty And Significance:** 3
**Empirical Novelty And Significance:** 2
**Recommendation:** 5

**Clarity, Quality, Novelty And Reproducibility:**

The structure of the paper is clear, but it will help if the authors could further clarify certain details in the algorithm design and emphasize the design of the merging rule of subclusters. It will also help if the authors could include a small-sized example dataset and show how the algorithm works. Figure 2 seems confusing to me.
For example there are some minor issues:
1) On page 3, the second paragraph of section 3, I think the notation E is used before it is formally defined?
2) Same thing applies to "mutually visible boundary". It might be better to remind readers of the definition.
3) The authors can also emphasize certain definitions such as maximal black nodes to make the algorithm description more clear.

**Strength And Weaknesses:**

Strengths:

- I like the idea of relying on growing and merging high-dimensional hyperrectangles in Euclidean space to speed up agglomerative hc. This seems to be a natural approach that is worth studying.
- The algorithm returns reasonable results on the datasets tested. The clustering quality and efficiency seem to be promising, if not sufficient.

Weaknesses:
- After taking a closer look, the partition of the whole space into small blocks and construction of the hypertree representation based on mesh cells looks similar to rescaling and dicretizing the different dimensions of the data points. The agglomerative algorithm built on top of that looks like a generalization of single-linkage, or MST rule (iteratively choosing two clusters with smallest shortest distance between any two points cross-clusters ) for mesh cells. In that sense, I wonder if the algorithm is significantly or only moderately different from existing approaches. I think it might help if the authors could emphasize this aspect.
- The idea is somewhat similar to a recent line of existing work, which constructs hc trees efficiently by geographically constructing local buckets or blocks that contain only very few points and merging on top of that (add citations) using Approximate Nearest Neighbor techniques. This line of work is not mentioned in the paper. In fact the community has witnessed sub-quadratic run time bounds on general agglomerative HC algorithms using this approach. See for example the following papers: Abboud, Amir, Vincent Cohen-Addad, and Hussein Houdrougé. "Subquadratic high-dimensional hierarchical clustering." Advances in Neural Information Processing Systems 32 (2019); Moseley, Benjamin, Sergei Vassilvtiskii, and Yuyan Wang. "Hierarchical clustering in general metric spaces using approximate nearest neighbors." International Conference on Artificial Intelligence and Statistics. PMLR, 2021.
- The experiment design is not very thorough. The datasets used seem to be limited: a few datasets with similar cluster constructions, and low-dimensional.  I would be interested in seeing how the algorithm performs on high-dimensional datasets to see if the tree construction grows fast as D becomes big. Further, since the paper focuses on hierarchical clustering, the baseline should also include other hc approaches such as single- average- and centroid-linkage.

**Summary Of The Paper:**

This paper focuses on designing a hierarchical clustering (hc) algorithm with improved efficiency. The algorithm, named DAC (divide-and-conquer), detects local clusters in small neighborhoods and then merge them in a hierarchical order. The algorithm starts with a hyperrectangle (block) that contains all the data points, and iteratively dissecting the current hyperrectangle into 2^D smaller hyperrectangles and thus creating a hypertree. The dissection of blocks continues until every point has found its own block (mesh cells). The algorithm then adopts a bottom-up approach to merge the blocks. The method is tested on several datasets. The algorithm is tested on several datasets and compared with general clustering baselines.

**Summary Of The Review:**

Overall this paper contains some interesting ideas, which could potentially give rise to efficient hc algorithms indeed. However, after reading the paper, I feel the design of the hypertree and the rule of merging maximal black nodes, even when combined together, does not really help us go beyond the dilemma of agglomerative hc algorithms being very slow. Rather than a new agglomerative algorithm with novel merging rules, the approach is closer to a variant of the MST, or single-linkage, algorithm. To that end, the experiment design does not seem to be sufficiently convincing either. It might not be ready for this venue yet.

---

> ### Author Response · Authors · 2022-11-19
> **Response to R1**
>
> We thank R1 for their helpful comments.
>
> 1 Rescaling and Discretizing: Yes, DAC does discretize each dimension of the space containing the points. It does so hierarchically, to get recursive tessellation of the overall (universe) hypercube of edge length E, containing all points, down to the minimum required box (mesh cell) size (highest required resolution). The lowest hypercube could be a unit hypercube (corresponding to a log at the minimum or larger if more than one points are allowed in it. The depth of the tree could thus be log 2^^E or less. This tree enables access to a point independent of the number of points, dependent only on E. Although it is a simple implementation of the Divide-and-Conquer principle, we are not aware of any algorithm that takes advantage of the principle for recursive clustering.
> The agglomerative algorithm does  use single linkage/MST rule, but it applies it to components (clusters) of points, thus repeatedly merging clusters and not points. Further, since the merger is between the nearest clusters, the distance between two clusters is not computed between all pairs of points across clusters but only between pairs of points lying on the “surfaces” of the two clusters, and even on the surface, between the mutually exposed (visible) parts of the surface. This reduces the distance computation complexity to points on parts of the cluster surfaces, not to all pairs of points across clusters. Fig 1 describes this visibility based geometric aspect of the algorithm.
> To our knowledge, in both these aspects our algorithm is unique among the clustering algorithms in the literature.
>
> 2 Relationship to Approximate Algorithms: The local bucket algorithms either create buckets in a computationally expensive way or they process the buckets less efficiently or accurately than DAC. Abboud et al use MST and compute HC by repeatedly merging closest clusters as we do. We agree that approximation algorithms can improve the complexity of MST computation --  a trade off between the accuracy of the MST obtained (hence the quality of HC) with the complexity of the MST computation  -- and they reduce MST computation from quadratic in the number of points to subquadratic by an amount proportional to the degree of the approximation error allowed. These approximation algorithms also consider closest average, and average squared, distances between pairs of points across clusters, as well as metrics that are simpler than the Euclidean distance. However, the objective of these algorithms is an approximate result. DAC instead is aimed at obtaining cluster hierarchy by merging truly closest clusters.
>
> 3 DAC Properties: We have presented DAC's complexity in terms of Big O notation by enumerating all component complexities. We have also in discussed the problem of testing the quality of clustering on high-dimensional datasets our experiments with it. Since there is no visualization of high dimensions possible without projection to 2D or may be 3D space, it is not possible to verify the clustering performance visually. We have therefore divided the testing of the algorithm in two parts. First, we have tested the correctness of our D-dimensional DAC for the special case of D=2 and 3 because the correctness of the results is easy to verify visually. We have done so on 2D and 3D datasets that are used by almost all clustering algorithms for performance comparison with SOTA. The correctness of results here shows that our DAC results match the ground truth and with other algorithms. The second part is about testing the algorithm for D>3. This is necessary since our algorithm is meant to have a better complexity as a function of D. We have done so in two ways. We have tested the quality of DAC clusters in 512 dimensional space and matched the clusters with those obtained by a CNN classifier. The results are a nearly complete match. Second, we have empirically estimated the run times of DAC against other algorithms representative of different past approaches, for varying values of D and the number of points. We see that DAC has the slowest increase with n.
>
> 4 Definitions: We have defined E, and added a figure (Fig 1) to geometrically illustrate steps in the algorithm, including definitions of mutually visible boundary and maximal black nodes.
>
> 5 Novelty: The reasons we think DAC is new are: (a) It decomposes the point set into parts independent of the number and distribution of points n (therefore computationally efficient). (b) The recursive merger between cluster pairs is indeed done using a like single linkage like (MST) algorithm; however, the single links are not identified by considering all pairs of points across the two clusters; rather, they are between points on a part of the “surface” or boundary of each of the clusters. Fig 1 gives an overview of DAC that may help in bringing out its novelty. Experimental validation of DAC properties is discussed above.
>
> Again, we thank R1 for getting us to think harder.

---

### Author Response · Authors · 2022-11-19
**Response to All Reviewers**

We thank reviewers for their valuable comments. The comments have helped us learn and present more in the updated submission.

Our responses to individual review comments are in the same order as the comments in the reviews.

We would like to continue adding references as we encounter additional relevant work, including those suggested by the reviews.

We welcome additional suggestions.

Thank you.

---

### Decision · Program_Chairs · 2023-01-20

**Decision:**

Reject

**Justification For Why Not Higher Score:**

The weaknesses of the paper that I have listed above are crucial and should be addressed for publication of this paper. I strongly recommend addressing these issues, that is, experiments on high-dimensional data, discussion and comparison with related space-partitioning methods, and revision of presentation.


**Justification For Why Not Lower Score:**

N/A

**Metareview: Summary, Strengths And Weaknesses:**

This paper proposes a two-step procedure for spatial clustering. It first performs recursive discretization of data points to divide them into smaller blocks, followed by merging them into hierarchy of clusters, which naturally leads to a tree structure of clusters. The efficiency and the effectiveness of the proposed method is empirically evaluated on several real-world datasets.

### Strength
- The proposed method is empirically shown to be more efficient than CLIQUE, which is the popular mesh-based clustering algorithm.

### Weakness
- Experiments are not thorough and the performance of the proposed algorithm is not convincing at the moment. I imagine that the effectiveness of such discretization-based methods quickly vanishes if the dimensionality of data gets higher due to the curse of dimensionality; that is, every data point is immediately isolated into a single block at the first couple of divisions. If this is not the case of the proposal, the authors should show it theoretically and/or empirically. Unfortunately, all datasets used in the experiments are low-dimensional. Although the authors argue that it is difficult to visualize results for high-dimensional data, it is not problematic. It is enough to present the resulting quality score like NMI for various datasets, and some dimensionality reduction methods like tSNE or UMAP can be used for visualization.

- As reviewers pointed out, the advantage of efficiency is questionable. For example, KD-tree is popularly used with DBSCAN, which can speed-up the algorithm especially for low-dimensional data. The advantage of the proposal compared to such space-partitioning algorithms is not clear at the moment.

- Presentation should be improved. There are inaccurate sentences as reviewers pointed out. I also recommend revising plots, as fonts are too small and hard to read.

Overall, the current manuscript is not ready for publication.